# Combustion Synthesis and Reactive Spark Plasma Sintering of Non-Equiatomic CoAl-Based High Entropy Intermetallics

**DOI:** 10.3390/ma16041490

**Published:** 2023-02-10

**Authors:** Kirill Vasilevich Kuskov, Andrey A. Nepapushev, Sofiya Aydinyan, Dmitry G. Shaysultanov, Nikita D. Stepanov, Khachik Nazaretyan, Suren Kharatyan, Elena V. Zakharova, Dmitry S. Belov, Dmitry O. Moskovskikh

**Affiliations:** 1Center of Functional Nano-Ceramics, National University of Science and Technology MISiS, 119049 Moscow, Russia; 2Department of Mechanical and Industrial Engineering, Tallinn University of Technology, 19086 Tallinn, Estonia; 3Laboratory of Macrokinetics of Solid State Reactions, A.B. Nalbandyan Institute of Chemical Physics, Yerevan 0014, Armenia; 4Laboratory of Bulk Nanostructured Materials, Belgorod State University, 308015 Belgorod, Russia; 5Department of Functional Nanosystems and High-Temperature Materials, National University of Science and Technology MISiS, 119049 Moscow, Russia; 6Research Laboratory of Scanning Probe Microscopy, Moscow Polytechnic University, 107023 Moscow, Russia

**Keywords:** intermetallics, high-entropy materials, mechanical activation, high-energy ball milling, sintering, mechanical properties, microstructure

## Abstract

The present work reports the direct production of a high-entropy (HE) intermetallic CoNi_0.3_Fe_0.3_Cr_0.15_Al material with a B2 structure from mechanically activated elemental powder mixtures. Fast and efficient combustion synthesis (CS), spark plasma sintering (SPS), and reactive SPS (RSPS) methods were used to synthesize the HE powders and bulks. The formation of the main B2 phase along with some amounts of secondary BCC and FCC phases are reported, and L12 intermetallic (CS scheme) and BCC based on Cr (CS + SPS and RSPS schemes at 1000 °C) were observed in all samples. The interaction between the components during heating to 1600 °C of the mechanically activated mixtures and CS powders has been studied. It has been shown that the formation of the CoNi_0.3_Fe_0.3_Cr_0.15_Al phase occurs at 1370 °C through the formation of intermediate intermetallic phases (Al_9_Me_2_, AlCo, AlNi_3_) and their solid solutions, which coincidences well with thermodynamic calculations and solubility diagrams. Compression tests at room and elevated temperatures showed that the alloy obtained by the RSPS method has enhanced mechanical properties (σ_p_ = 2.79 GPa, σ_0.2_ = 1.82 GPa, ε = 11.5% at 400 °C) that surpass many known alloys in this system. High mechanical properties at elevated temperatures are provided by the B2 ordered phase due to the presence of impurity atoms and defects in the lattice.

## 1. Introduction

Al-Co-Cr-Fe-Ni is one of the most popular systems based on which numerous high-entropy materials (HEM) have been created [1,2,3,4,5,6,7,8,9,10,11,12,13,14]. The initial studies were aimed at obtaining alloys with maximum configurational entropy with multiple principal elements in equimolar or near-equimolar ratios. It was expected that such an approach would lead to the formation of a single-phase solid solution rather than a set of intermetallic phases. It was assumed that the maximum entropy is the most significant parameter in the design of multicomponent alloys with improved characteristics. Further studies of these materials demonstrated that deviation from the equiatomic ratio of elements also makes it possible to obtain single-phase solid solutions. Furthermore, for example, it is possible to vary the phase composition in the already mentioned Al-Co-Cr-Fe-Ni system by varying the Al content. Alloys with FCC lattice are usually formed in this system at low Al concentrations [3,4,5,10,11,14]. Intermetallic phases with the B2 structure [2,3,4,15], which is an ordered BCC solid solution with the Pm3m space group, i.e., representing two conjugated primitive lattices of different metals, can form in this system at a high Al concentration. Changing the phase composition accordingly results in a change in the properties of the material. Numerous studies have demonstrated that BCC high-entropy alloys (HEA) have high-yield strength but low tensile ductility, while FCC HEAs generally have high tensile ductility but low yield strength. Recent works have proposed approaches to the production of non-equiatomic HEAs with a single, double, or multiphase structure, which makes it possible to increase the strength–ductility ratio. For example, changing the composition towards the nickel-rich region (AlCoCrFeNi_2.1_) allows achieving simultaneous strength–ductility enhancement [16]. Extremely high strengths have been achieved in multiphase systems due to various strengthening mechanisms, such as particle hardening [17], precipitation hardening at aging [18] or solid solution strengthening [19]. In the latter case, the BCC phase with B2 precipitation in AlCoCrFeNi had an ultra-high hardness of 1124 Hv due to the enhanced solid solution strengthening effect. Thus, in non-equiatomic alloys, properties superior to those of equiatomic alloys may be achieved. Furthermore, the use of non-equiatomic HEAs makes it possible to significantly expand the compositional space for study.

Most HEAs are obtained by different melting technics [2,3,5,8,10,11,12] or mechanical alloying [4,6,7,9,20,21]. These methods allow destroying the crystalline structure of the starting materials and effectively mix the components. Crystallization from the melt allows utilization in standard metallurgical furnaces, producing material in the form of a ready-to-use ingots. Mechanical alloying has high performance and efficiency, as well as the ability to combine metals with different properties. The common shortcoming for both processes is their duration, which may reach several hours in order to ensure better homogenization. For most of the melted HEA, subsequent annealing is necessary, while mechanical alloying may lead to powder contamination by a lining material of the milling jars, and consolidation is required for the production of bulk materials [22]. The alternative method to produce HEAs in a powder form may be combustion synthesis or self-propagating high-temperature synthesis. The method employs the use of heat from exothermic chemical reactions occurring in the wave, which propagates through a sample during combustion synthesis or in the thermal explosion mode to obtain various chemical compounds. High wave propagation velocities (0.1–20 cm/s) ensure high process productivity; at the same time, there is no need for long-term high-temperature external heating, since the process occurs in a self-sustaining mode. In the case of HEA, direct combustion synthesis from elements is difficult due to the small amount of heat released during the fusion of elemental metals [22]. Therefore, a thermite-type reaction or a thermally coupled reaction accompanied by a large release of heat sufficient to melt metal components and form HEA can be used [22]. However, the latter approach is mainly suitable for the production of cermets, in which HEA is used as a metallic binder for ceramic grains [23,24].

One way to enhance the reactivity of the powder mixture is to activate it by treatment in various apparatuses, mainly planetary mills [25]. The main difference from the above-mentioned mechanical alloying approach is that mechanical activation (MA) does not last as long and does not lead to the formation of new phases. During MA, the powders are crushed and the surfaces of the particles are cleaned from the oxide films. The contact area between components increases. Often, it allows the decreasing of the initiation temperature and promotes a more complete reaction in mixtures that burn weakly without activation [26]. Since Al-Co-Cr-Fe-Ni refers to systems in which SHS is difficult or impossible (depending on the composition), MA can be used to initiate a reaction in them without the use of a thermally coupled reaction. Moreover, preliminary activated powders can be used not only for combustion synthesis, but also for reaction sintering, which is based on the idea of simultaneous synthesis and compaction of the material. This method produces a wide range of materials: composites [27,28,29,30], thermoelectrics [31,32,33,34], and ceramics [35,36,37], including high-entropy ones [38,39,40,41]. The implementation of this method is typically carried out on hot pressing (HP) and spark plasma sintering (SPS) devices and referred to as RHP and RSPS, respectively [42].

SPS is widely used to produce HEM in the Al-Co-Cr-Fe-Ni system [43,44,45,46]. The majority of publications are dedicated to the production of HEA from mechanically alloyed powders. In this case, the main formation of the alloy occurs at the stage of mechanical treatment in a mill, while SPS is used mostly for homogenization and compaction. In contrast, during RSPS, the heating of the mechanically activated mixture causes chemical reactions to occur, enhancing the sintering process. Recent investigation into the (RSPS) process [47] shows that using CS directly during the SPS provides better density and mechanical properties of the material compared to separate sequential use of the MA and SPS methods. In addition, the combination of CS and RSPS allows a significant reduction in the production time of the bulk material.

Thus, this work had several goals: (i) to directly synthesize non-equiatomic HEA powders in an Al-Co-Cr-Fe-Ni system from mechanically activated mixtures by fast and efficient CS; (ii) to produce bulk HEA by the RSPS method; and (iii) to compare the mechanical properties of the samples made by sintering the CS powder and by reaction sintering on an SPS setup.

## 2. Materials and Methods

Elemental powders (all from Russian manufactures) of Al (>98.6% purity), Co (99.8%), Fe (99.8%), Ni (99.9%), and Cr (>99.1% purity) were used as initial materials; 35 at.% Al + 35 at.% Co + 10 at.% Ni + 10 at.% Fe + 10 at.% Cr. Mechanical activation (MA) was carried out in the Activator-2S high-energy planetary ball mill (Chemical Engineering Plant Ltd., Novosibirsk, Russia,). The sunwheel rotation speed was 694 rpm, and the ratio between rotation speeds of the sunwheel and the jars was 1 (K = 1.0). Ball-to-powder mass ratio was 20:1 (360 g stainless steel balls: 18 g of powder). Argon at a pressure of 0.4 MPa was used as a protective gas inside the jars; treatment duration varied from 1 to 5 min. Additional 3 min of treatment with 10 mL of hexane was performed to increase the powder yield.

Combustion syntheses were conducted in a pressure reactor at 0.1 MPa argon atmosphere, and reaction in a powder mixture was initiated by a tungsten coil. The combustion parameters were studied by high-speed video recording on a pressed 10 mm in diameter and 14–15 mm high cylindrical samples, the relative density of which was ~55%.

Thermodynamic calculations were performed by “Thermo” package (ISMAN, Chernogolovka, Russia) [48]. The equilibrium products’ composition and combustion temperatures for a mixture of various elements were calculated by minimization of the thermodynamic potential under the assumption of the adiabatic process, which proceeds without heat losses from the reaction zone [49]. “Thermo” considers only chemical reactions and phase transitions of known reagents and products, without considering formation of solid solutions and multicomponent compounds, when calculating combustion products and parameters.

Spark plasma sintering was implemented in vacuum at a Labox 650 unit (Sinter Land, Nagaoka, Japan) at 50 MPa of applied pressure with a dwell time of 10 min. The reaction SPS (referred to as RSPS in the manuscript) for mechanically activated mixtures was carried out at a heating rate of 200 °C/min, and the combustion products (referred as CS + SPS) were sintered at 100 °C/min.

The density of sintered materials was determined by the hydrostatic weighing method (Archimedes method), and the theoretical density was calculated according to the additivity rule.

X-ray diffraction patterns were obtained using a Difray-401 setup (JSC Scientific Instruments, St.-Petersburg, Russia) using Cr radiation. The diffraction patterns were processed using the Jade software.

Electron microscopy was carried out on Vega (Tescan, Brno, Czech Republic) and FSM 7600 (JEOL, Tokyo, Japan) instruments using backscattered electron detectors and attachments for EDS analysis (Oxford Instruments, Abingdon, UK).

Differential scanning calorimetry analysis was carried out on STA 449 F1 Jupiter equipment (Netzsch, Selb, Germany) with a heating rate of 20°/min to 1600 °C in an Ar flow (purity 99.999%).

The microhardness of the bulk specimens was measured using Vickers hardness tests with a DuraScan 70 (Emco-Test, Kuchl, Austria) under applied loads of 4.9 N.

The mechanical properties were evaluated on rectangular specimens of 4 × 4 × 6 mm^3^ in size at room temperature and at 400 and 600 °C using a test machine 300 LX (Instron, Norwood, MA, USA) equipped with a radial furnace. The specimens were placed into the furnace, preheated to the testing temperature, and held for ~10 min to equilibrate the temperature before testing. The temperature was controlled by a thermocouple attached to a side surface of the specimen. The initial strain rate was 10^−4^ s^−1^.

## 3. Results and Discussion

### 3.1. Combustion Synthesis of the High-Entropy Intermetallic Compound

To produce reactive particles, MA of the powder mixtures was carried out in a high-energy planetary ball mill for 1–5 min. The minimum treatment duration, at which no new phases were formed in the jars and combustion of the blend in the reactor proceeded in a self-sustained mode, was 3 min. Further increase of the processing time up to 5 min had not led to a change in the temperature and burning rate, so this mode was set as optimal.

The powder diffraction pattern of the mechanically activated mixture (Figure 1) presents only peaks of the initial metals. After the combustion synthesis picture changes, one can observe the main B2 phase on the diffraction pattern as well as peaks of the secondary BCC, FCC phases, and the L1_2_ intermetallic phase. The lattice parameters were calculated by modeling the experimental peaks using OriginPro (v8.1.34.90) software by OriginLab Corporation (Northampton, MA, USA). The lattice parameters of these phases are given in Table 1.

The VEC value for the Al_3.5_Co_3.5_NiFeCr composition is 6.6, which should correspond to the formation of BCC structures [15,50]. According to the (FeCrNi)-Co-Al ternary diagram calculated using the Pandat software package [15], the considered composition at 1000 °C falls into the single-phase B2 region near to the two-phase BCC + B2 area. Comparing this diagram with those for the (FeCrCo)-Ni-Al system, it can be found that the B2 single-phase region existence is even smaller. Additionally, in [15], the Al_x_(CoCrFeNi) system in the concentration range of aluminum from 0 to 60 mol.% is considered, in which a wide two-phase (BCC + B2) region is represented. Based on the above data, we can assume the formation of both a single-phase and a two-phase structure with BCC lattice in our system. According to the calculation from our previous work [51], ordered B2 phases are thermodynamically stable in a system with a non-equiatomic composition, especially at a low Cr content. The presence of a local increase in the concentrations of such metals as cobalt and chromium also confirms the possibility of the L1_2_ and σ phases formation at sufficiently low temperatures. If we consider the process with respect to the combustion of the Al-Co system, the addition of Ni, Fe and Cr dilutes the reaction mixture and leads to a decrease in the combustion temperature and combustion rate due to energy consumed in the heating of non-reacted elements. Under the conditions of a real experiment, the system comes to an intermediate result between calculations in Thermo and Pandat.

The microstructure and EDS element map of particles before CS is shown in Figure 2. Intensive mechanical treatment leads to the formation of layered composite particles, the thickness of the structural components of which is within a few micrometers. Such a characteristic structure usually forms during the joint processing of metals with different properties: brittle ones (Co, Ni, Cr) split into small pieces, and ductile ones (Al, Fe) deform and flatten, becoming a matrix for brittle ones. At the same time, as already mentioned, the contact surface area between metals increases significantly, which leads to an increase in their reactivity and synthesis can be implemented in the mixture.

The microstructure of the synthesized powders is shown in Figure 3. A preliminary thermodynamic calculation in the Thermo program shows that the combustion temperature of the 3.5 Al + 3.5 Co + Ni + Fe + Cr mixture is 1377 °C, and the products composition is 60 wt.% AlCo, 14 wt.% AlNi_3_, 4 wt.% Co, 11 wt.% Cr, 11 wt.% Fe. Typically, measured combustion temperature is lower than that calculated in the program, since in the latter case it is calculated under adiabatic conditions (without considering heat losses and not considering solid solution or secondary intermetallics formation). The distribution of elements confirms the calculated composition (Figure 3). A slight intersection of the chromium and iron regions is observed, so it safe to assume that heating stimulated the formation of solid solutions based on iron and/or chromium during the reaction.

Summarizing the results of XRD (Figure 1), SEM + EDS (Figure 3) and the abovementioned calculations, the formation of a multiphase structure after CS is reasonable. The basis is an intermetallic AlCo (B2) phase (dark grey on Figure 3); also present are Fe and Cr phases with small areas of their mutual dissolution (BCC, light regions), AlNi_3_ intermetallic compound (L1_2_, light grey), and some amount of Ni-Fe solution (FCC). Elemental chemical composition is presented in Table 2. It can be concluded that mechanical treatment in a planetary mill led to a relatively uniform distribution of components in the system, but part of the nickel that had no contact with aluminum formed a solid solution with iron or cobalt [51], leading to a more complete AlCo formation reaction.

To reveal the possible interaction mechanism between the components during heating, DSC studies were carried out. According to the obtained data, the formation of intermetallic phases occurs in several stages (Figure 4), which is also confirmed by studies in binary Al-Ni and Al-Co systems [52,53]. Therefore, a first peak at ~264 °C (Figure 4), a solid solution and intermetallic phases of the Al_9_Me_2_ type are formed (Me = Ni, Co), then further reactions occur until the formation of AlCo and AlNi_3_ (broad peak at 450 °C).

Further dissolution of the components occurs along with the heating, as evidenced by a wide endothermic peak on the DSC curves for the MA and CS powders. The heat flow curve changes its direction towards exothermic processes in the temperature range of 1200–1300 °C. The increase in the thermal effect is probably associated with the completion of the solid solutions’ formation. A similar effect was observed in this system with the copper addition [4]. However, on the exothermic section of the curve, endothermic peaks exist at 1370 °C and 1485 °C. These temperatures do not correspond to the melting temperatures of binary compounds possible in this system [54].

ThermoCalc calculations of high-entropy intermetallic compounds (Fe_0.25_Co_0.25_Ni_0.25_Cr_0.25_)Al and (Fe_0.2_Co_0.2_Ni_0.2_Cr_0.2_)Al [55] predict the appearance of a liquid phase at 1506 °C and 1450 °C, respectively. At the same time, in [15] a simulated phase diagram in the Al_x_CoNiFeCr system shows 1420 °C and 1506 °C maximum temperatures for solidus and liquidus, respectively. When deviating from the equiatomic composition, despite the discrepancies in theoretical calculations, it can be stated with confidence that the broad endothermic peak on the DSC curve at 1485 °C corresponds to the melting of the high-entropy CoNi_0.3_Fe_0.3_Cr_0.15_Al intermetallic compound.

### 3.2. Spark Plasma Sintering of the High-Entropy Intermetallic Compound

The behavior of reactive mixtures under high-speed heating rates [51] and results of the reaction sintering in the Ni-Al system [47] showed the absence of a pronounced self-sustaining reaction at low heating rates. Based on that, a heating rate of 200 °C/min was chosen for the RSPS of the reactive particles. In turn, since a lower heating rate allows for better density [47], the powders after CS were SPSed at 100 °C/min. The starting sintering temperature of 800 °C was chosen based on the ThermoCalc modeling data, which indicated 746 °C as the lower temperature for a single phase compound formation [51]. Table 3 shows the relative densities values and hardness obtained after sintering.

X-ray diffraction patterns after consolidation mainly show the same phases as after CS (Figure 5). At 800 °C sintering temperature, B2, L1_2_, BCC and σ-phases exist at samples. The presence of the σ phase indicates the ongoing dissolution of iron and chromium into each other with the heating of the material. The temperature increase up to 1000 °C leads to the disappearance of this phase, the upper existence limit of which is 856 °C [54]. At further temperature increase, carbide and oxide impurity phases form. Phase compositions at different sintering temperatures are gathered in Table 4.

Sintering at different temperatures confirms our conclusions after DSC analysis. At 800 °C, the X-ray diffraction pattern (Figure 5) retains the intermetallic phases (B2 and L1_2_), BCC and the σ phase in the structure, and the FCC phase disappears or merges with L1_2_. With an increase in the sintering temperature to 1000 °C, the σ-phase decomposes and the components further dissolve in each other, leading to a uniform distribution in the structure, except for Cr. We can suppose that a CoNi_0.3_Fe_0.3_Cr_0.15_Al high-entropy intermetallic compound was formed (the composition was determined based on the EDS analysis) [55]; moreover, dissolving chromium can occupy a place both in the aluminum sublattice and in the Co sublattice [56]. It is interesting to note that, according to EDS data, the amount of Cr transferred into the multicomponent phase was about 50%.

The microstructure of the sintered samples is presented in Figure 6. As mentioned above, a comparison of XRD data of sintered by CS + SPS and RSPS materials did not reveal any significant differences in the phase composition. However, SEM images showed that, in the RSPS samples obtained at 800 °C, chromium-rich light inclusions had blurred boundaries (Figure 6a), indicating an interaction with the surrounding particles. As the temperature increased to 1000 °C, the difference in microstructures became more noticeable: chromium inclusions had a smaller size and were more uniformly distributed. This difference can be explained by the sintering conditions. According to the CS + SPS scheme, the powder was first synthesized in a reactor. Combustion initiated by a heated coil proceeded in an oscillatory mode, i.e., the propagation of the reaction front was unstable, indicating the difficulty in the components’ interaction in the mixture. As a result, a porous matrix was formed from a more refractory intermetallic compound with inclusions of individual metals. In the CS + SPS process, the powder die was subjected to a constant load of 50 MPa, which resulted in closer interparticle contacts and contacts between different metals. Consolidation of the material began before the formation of a refractory intermetallic compound. Local overheating occurred at the point of particle contact during pulsed heating under SPS conditions [57], as well as uniform heating of the entire volume due to Joule heat. Thus, the reaction was initiated in the entire sample’s volume. In addition, due to constant heating, the temperature gradient between the points where the exothermic reaction occurred and the rest, powder volume decreased. All these factors led to the formation of a more homogeneous structure during the RSPS process.

### 3.3. Mechanical Properties

Figure 7 shows the results of mechanical compression tests of RSPS (a) and CS + SPS (b) specimens sintered at 1000 °C. The values of yield strength (σ_0.2_), peak strength (σ_p_) and relative strain (ε) at different test temperatures are presented in Table 5. At room temperature, the CS + SPS sample is brittle because the values of its yield strength and tensile strength differ insignificantly—2.48 and 2.49 GPa, respectively, and the fracture strain was 0.2%. The RSPS sample exhibits some plastic properties at room temperature: σ_0.2_ = 2.30 GPa, σ_p_ = 2.58 GPa, ε = 3.5%. It can also be seen that the elastic properties of these materials are approximately on the same level. This level of mechanical properties surpasses lots of known alloys in this system.

With temperature increase, the plastic properties of materials varied to a different extent. At 400 °C, the yield strength and relative strain were 1.82 GPa and 11.5% for RSPS and 1.40 GPa and 1% for CS + SPS specimens. Both samples showed a decrease in yield strength, but tensile strength changed in a different way: for CS + SPS it decreased to 1.54 GPa, and it increased to 2.79 GPa for RSPS.

This level of properties is determined by the structure consisting of BCC phases: a high entropy ordered B2 matrix phase and coherent inclusions of the BCC phase based on chromium. Thus, the main contribution to the mechanical properties at room temperature was made by ordering and coherent hardening. The preservation of high mechanical properties at elevated temperatures was also due to the presence of a B2 ordered phase. The mechanical properties of the B2 structure were comprehensively studied [58]. It was noted that such materials have several slip systems, and in the B2 structures slip predominantly occurs along the <100> (three independent slip systems) or <111> (five independent slip systems) directions. Polycrystals of AgMg and CuZn (and compounds exhibiting (100) slip such as NiAl) are most ductile where they exhibit the lowest yield strength, i.e., at stoichiometric composition. In contrast, FeAl is brittle, fractures intergranularly and has the highest yield strength at a stoichiometric composition. The plasticity of FeAl increases and the fracture mode transfers to transgranular with an increase in the Fe content. Thus, in our Al_3.5_Co_3.5_NiFeCr composition, a nonstoichiometric intermetallic compound CoNi_0.3_Fe_0.3_Cr_0.15_Al is presumably formed, which has signs of sliding both in the direction [100] (the highest plasticity at the minimum yield strength) and in the direction [111] (the demonstration of plastic properties at room temperature with a deviation from stoichiometry).

A further temperature increase repeats the patterns previously identified for the CS + SPS sample—a decrease in σ_0.2_ (1.12 GPa) and σ_p_ (1.38 GPa) and an increase in ε up to 25%. The increase in plasticity with temperature increase significantly exceeded the drop in the tensile strength and yield strength values. It can be argued that the brittle-ductile transition temperature for this material is in the range of 400–600 °C. At 600 °C the relative strain of the sample exceeded 50%, which made it impossible to measure the tensile strength. The yield strength, as in previous cases, decreased to a 1.03 GPa.

It is known that B2 structures upon heating up to 0.4·T_m_ retain their strength properties. Impurity atoms under these conditions have a strong influence on the yield, which leads to noticeable yield strengths. A change in slip systems, which is observed in some compounds with temperature increase, for example FeAl, is also possible. Since there is still strain hardening at 600 degrees, it can be assumed that 0.4·T_m_ will be slightly less than 600 °C, which corresponds to the endothermic peak on DSC (0.4·(1370–1485) = 548–594 °C). Additionally, the presence of five elements in the composition leads to a large number of defects. Vacancies in non-stoichiometric alloys enhance diffusion, which leads to a decrease in the yield strength with increasing temperature.

Thus, the Al_3.5_Co_3.5_NiFeCr alloy obtained by the RSPS method demonstrated a good level of mechanical properties during compression (Table 4): a relatively high strength at room temperature, which remained and even slightly increased at temperatures up to 400 °C at the presence of plastic deformation, with pronounced strain hardening. At 600 °C a softening of the material was observed, but strain hardening was still preserved.

## 4. Conclusions

The high-entropy CoNi_0.3_Fe_0.3_Cr_0.15_Al powders were produced by the CS from the high-energy ball milled elemental component mixture. It has been demonstrated that a self-sustaining reaction can already be initiated in the mixture after 3 min of treatment due to the formation of composite reactive particles. An investigation of the combustion products revealed the formation of the B2 phase, secondary BCC, FCC, and the L1_2_ intermetallic phases. Based on the DSC measurements up to 1600 °C, a possible interaction mechanism during the heating of the milled powders was established, which consisted of the formation of intermediate solid solutions and intermetallic phases before the CoNi_0.3_Fe_0.3_Cr_0.15_Al phase formation.

Using mechanically activated powder and powder after the CS allowed for an easy and fast producing route for bulk HEA. For both the CS + SPS and RSPS schemes, sintering at 1000 °C for 10 min at 50 MPa led to the formation of dense (>90%) CoNi_0.3_Fe_0.3_Cr_0.15_Al high-entropy intermetallic compound. The total processing time required to produce the bulk sample (milling duration + sintering) was not in excess of 15 min. Mechanical tests indicated that the material exhibited an increase in plasticity while maintaining a high-yield strength at low temperatures. At elevated temperatures, a maximum tensile strength of σ_p_ = 2.79 GPa was achieved by the RSPS sample at 400 °C. This work demonstrates that combining high-energy ball milling and reactive sintering provides an efficient and fast fabricating route for obtaining high-performance structural materials.

## Figures and Tables

**Figure 1 materials-16-01490-f001:**
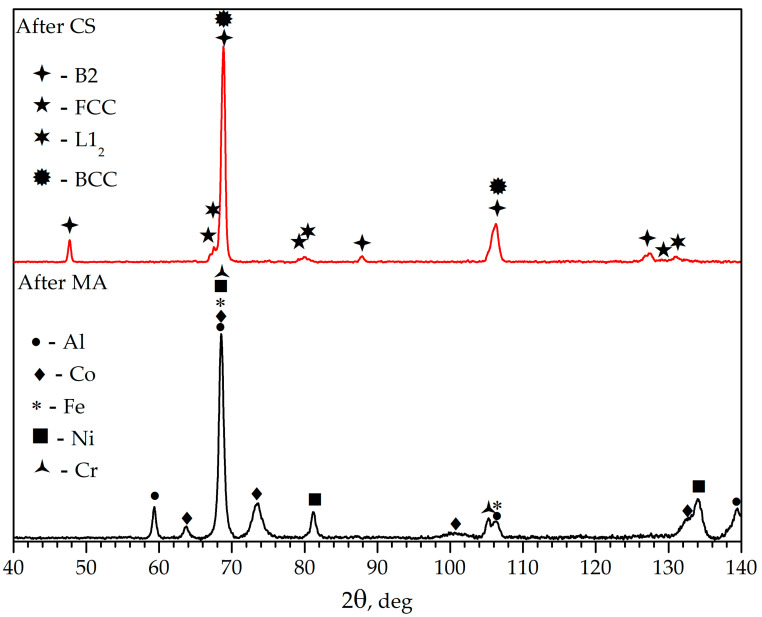
X-ray diffraction data for the MA and CS powders.

**Figure 2 materials-16-01490-f002:**
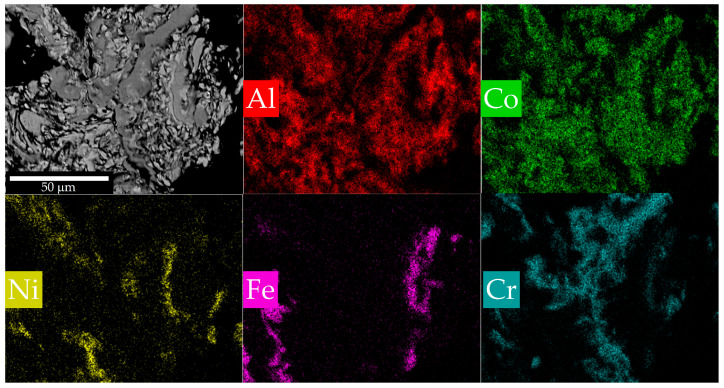
Cross-section and EDS element map of the powder after MA.

**Figure 3 materials-16-01490-f003:**
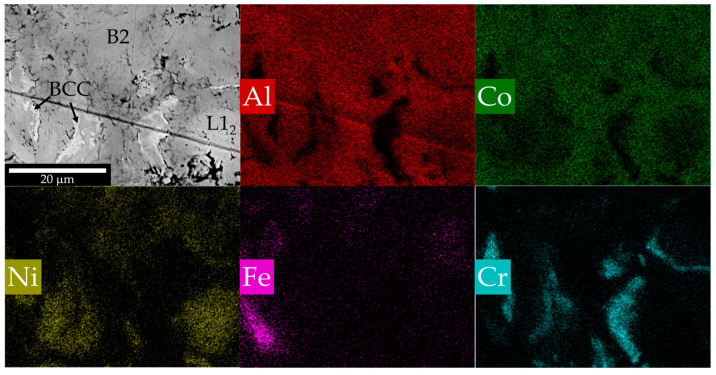
Cross-section and EDS element map of the powder after CS.

**Figure 4 materials-16-01490-f004:**
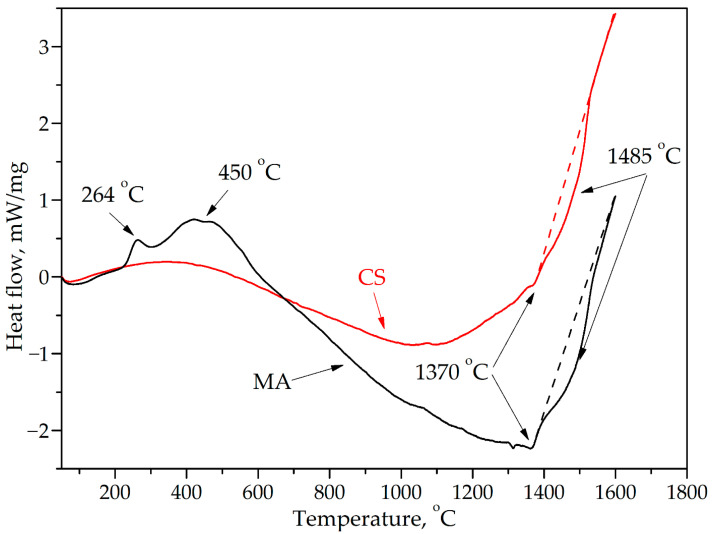
DSC curves for MA and CS powders.

**Figure 5 materials-16-01490-f005:**
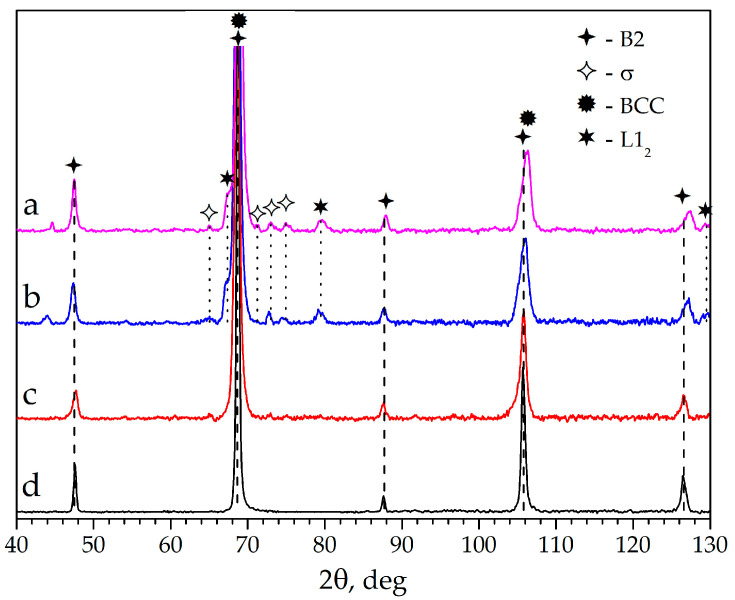
X-ray diffraction data for the sintered samples: CS + SPS at 800 (**a**) and 1000 °C (**c**), RSPS at 800 (**b**) and 1000 °C (**d**).

**Figure 6 materials-16-01490-f006:**
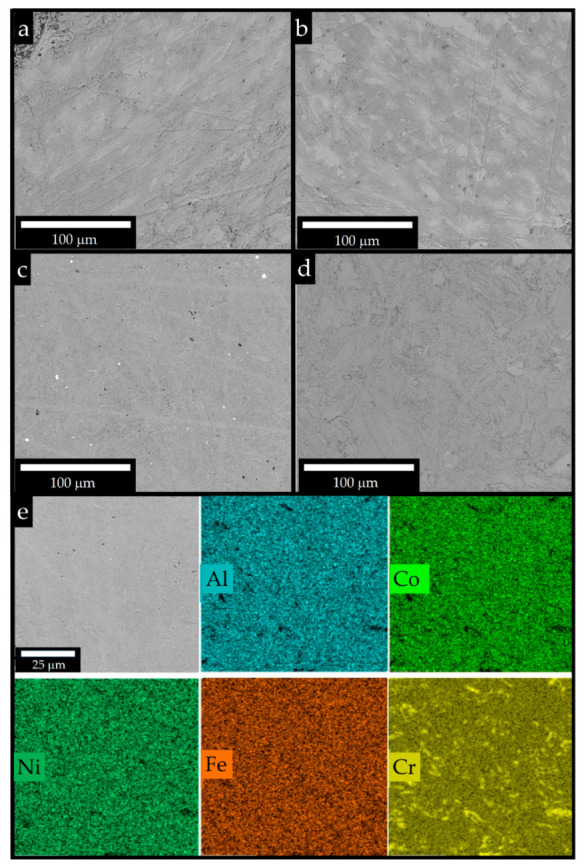
Microstructure of the samples sintered by RSPS at 800 °C (**a**) and 1000 °C (**c**), and CS + SPS at 800 °C (**b**) and 1000 °C (**d**), and EDS mapping RSPS at 1000 °C (**e**).

**Figure 7 materials-16-01490-f007:**
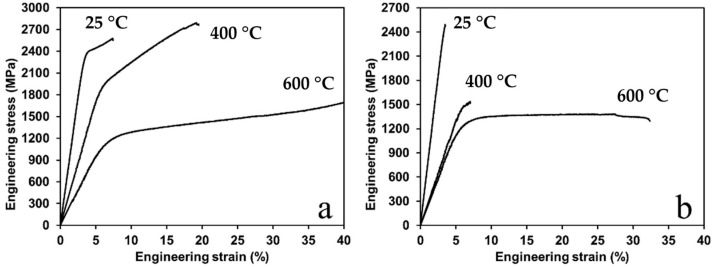
Stress–strain curves obtained during the compression tests of the Al_3.5_Cr_3.5_NiFeCr alloy after RSPS (**a**) and CS + SPS (**b**).

**Table 1 materials-16-01490-t001:** Notation and lattice parameters of the identified phases after CS.

Notation	Basis	Lattice Type	Lattice Parameter, Å	Lattice Parameter According to the ICCD Card, Å
B2	AlCo	Pm-3m	2.858	2.861 (#65-4903)
BCC	Fe/Cr	Im-3m	2.874	2.867 (#65-4899)/2.884 (#06-0694)
FCC	(Fe, Ni)	Fm-3m	3.591	3.598 (#47-1417)
L1_2_	AlNi_3_	Pm-3m	3.562	3.567 (#65-3245)

**Table 2 materials-16-01490-t002:** Chemical analysis of different phases after CS.

Phase	Al	Co	Ni	Fe	Cr
B2	36.2 ± 0.3	38.7 ± 0.2	11.2 ± 0.2	6.3 ± 0.2	7.6 ± 0.2
BCC	1.9 ± 0.3	0.9 ± 0.2	7.2 ± 0.2	10.9 ± 0.2	79.1 ± 0.2
	1.0 ± 0.3	2.1 ± 0.2	2.9 ± 0.2	88.3 ± 0.2	5.7 ± 0.2
L1_2_	33.6 ± 0.3	16.8 ± 0.2	46.9 ± 0.2	1.5 ± 0.2	1.2 ± 0.2

**Table 3 materials-16-01490-t003:** Relative density and hardness of samples after consolidation.

Sintering Method	Temperature, °C	Relative Density, %	Hardness, GPa
RSPS	800	79 ± 2	3.3 ± 1.0
	1000	97 ± 2	6.9 ± 0.7
CS + SPS	800	69 ± 2	3.0 ± 0.8
	1000	92 ± 2	6.7 ± 0.7

**Table 4 materials-16-01490-t004:** Phase composition of bulk materials after CS + SPS and RSPS, depending on the temperature.

Temperature, °C	Phases				
	Lattice			Basis	
800	B2	2.857	Pm-3m	AlCo	2.861
	L1_2_	3.582	Pm-3m	Al3Ni	3.567
	BCC	2.881	Im-3m	Cr	2.884
	σ	8.830/4.532	P42/mnm	FeCr	8.797/4.558
1000	B2	2.863	Pm-3m	AlCo	2.861
	BCC	2.887	Im-3m	Cr	2.884

**Table 5 materials-16-01490-t005:** Mechanical properties of Al_3.5_Co_3.5_NiFeCr alloy sintered at 1000 °C.

Sintering Method	Temperature, °C	έ, s^−1^	σ_0.2_, MPa	σ_p_, MPa	ε, %
RSPS	25	10^−3^	2295	2576	3.5
	400	10^−3^	1820	2790	11.5
	600	10^−3^	1030	–	>40
CS + SPS	25	10^−3^	2480	2491	0.2
	400	10^−3^	1400	1538	1
	600	10^−3^	1120	1380	25

## Data Availability

Not applicable.

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
