# Peer review of "Combustion Synthesis and Reactive Spark Plasma Sintering of Non-Equiatomic CoAl-Based High Entropy Intermetallics"

_materials, 2023, doi:10.3390/ma16041490_

Round 1

Reviewer 1 Report

The reviewed manuscript investigates combustion synthesis and reactive spark plasma sintering of non-equiatomic CoAl-based high entropy intermetallic. Authors demontrated that high-entropy material with intermetallic CoNi0.3Fe0.3Cr0.15Al matrix with B2 structure and Cr particles were produced from elemental powder mixtures by combustion synthesis (CS) and re-active spark plasma sintering (RSPS) methods.

1.    The description of the abstract should be specific. For examples, Line 23, a possible mechanism of the reaction was discussed. What is possible mechanism should be specific. Line 25, possible reasons for this behavior are also considered. What are possible reasons should be specific.

2.    The introduction needs a lot of enhancement. I think it needs to be extended to illustrate the novelty and the importance of the work, as well as some previous related studies in published literatures. This paper is not innovative enough. The highlights of this work are very common, no breakthrough.

3.    Line 95, powder material production companies need to be mentioned in Materials and Methods.

4.    On mechanical properties, Fig. 7 and Table 4 are confusing. For examples, relative strain 11.5 % for RSPS and 1 % for CS+SPS specimens. How do we know that the relative strain strain of the sample exceeded 50 %?

5.    The mechanism of some results is insufficient. A discussion section is required to improve the readability and clarity of the manuscript.

6.    The conclusion section is too wordy and needs to be concise.

Author Response

  1. Abstract was rewritten.
  2. Introduction was modified.

  3. Information was added.

  4. The relative strain was determined by drawing from the point of fracture a parallel line to the elastic strain region. Thus, from our point of view, Figure 7 and Table 4 are in good agreement. For the RSPS sample at 600 ℃, the value in Table 4 changed to 40 % in accordance with the given graph on Fig 7. However, we want to note that under the experimental conditions, the sample reached the maximum deformation of 50% without a break.

  5. We are forced to disagree with this remark, the article is divided into parts corresponding to individual stages of the study: combustion synthesis, spark plasma sintering, and the study of mechanical properties. The introduction of a separate discussion section, in which it will also be necessary to separately highlight the different stages of the study, in our opinion, will lead to even greater confusion in understanding the presented data.

  6. This section was modified.

Reviewer 2 Report

I have read the article "Combustion synthesis and reactive spark plasma sintering of non-equiatomic CoAl-based high entropy intermetallic". Based on my expertise, I would like to recommend this article for publication in Materials . Below are some comments on the article.

1) Please add the main ternary diagrams related to the identified phases that you refer to in the following sentence: "According to the (FeCrNi)-Co-Al ternary diagram calculated using the Pandat software package..."
2) If the authors have the results of pointwise EDS elemental chemcial composition, these analysis it should be added to confirm the presence of the main identified phases in the powder and in the sintered alloys.
3) Is it possible to calculate the fraction of the B2 phase that is responsible for "Preservation of high mechanical properties at elevated temperatures"?

Author Response

  1. The abovementioned phase diagrams were obtained in another work, a reference to which is presented in the article (15.Yang, S.; Lu, J.; Xing, F.; Zhang, L.; Zhong, Y., Revisit the VEC rule in high entropy alloys (HEAs) with high-throughput CALPHAD approach and its applications for material design-A case study with Al–Co–Cr–Fe–Ni system. Acta Materialia 2020, 192, 11-19.). Unfortunately, we unable to provide these data due to copyright law.
  2. Information was added where it was possible.

  3. The volume content of the B2 phase in the sample was approximately 97%.

Reviewer 3 Report

The paper entitled; "Combustion synthesis and reactive spark plasma sintering of non-equiatomic CoAl-based high entropy intermetallic" is an interesting article that contains exciting experimental results. The manuscript is well and comprehensively written. Therefore, it can be accepted in Materials.

1. What is the main question addressed by the research?

·         Production of a new HEA material from elemental powders in different atomic ratios and investigation of its properties.

2. Do you consider the topic original or relevant in the field? Does it address a specific gap in the field?

·         The topic is relevant in the field, and can partially fill a specific gap in that field.

3. What does it add to the subject area compared with other published material?

·         This question should be asked to the authors so that it can be explained comparatively.

4. What specific improvements should the authors consider regarding the methodology? What further controls should be considered?

·         The authors of the manuscript can add the answer to this question as a suggestion in the concluding part of the article.

5. Are the conclusions consistent with the evidence and arguments presented and do they address the main question posed?

·         It is suitable for me. However, it can be improved.

6. Are the references appropriate?

·         References are appropriate and sufficient.

7. Please include any additional comments on the tables and figures.

·         Related phases such as FCC, BCC, LB1 can be marked on SEM images and EDS data can be given in tabular form.

Author Response

3. For the first time, high-entropy intermetallic compounds were obtained using combustion synthesis and powder metallurgy methods both in powder and during spark plasma sintering. These materials have been investigated and characterized.

5. Conclusions section was modified.

7. Information was added.

Round 2

Reviewer 1 Report

Accept